# Built Environment and Health Behaviors: Deconstructing the Black Box of Interactions—A Review of Reviews

**DOI:** 10.3390/ijerph16081454

**Published:** 2019-04-24

**Authors:** Anne-Sophie Travert, Kristi Sidney Annerstedt, Meena Daivadanam

**Affiliations:** 1School of Public Affairs, Sciences Po, 75007 Paris, France; annesophie.travert@sciencespo.fr; 2Department of Public Health Sciences, Karolinska Institutet, 171 77 Stockholm, Sweden; kristi.sidney@ki.se; 3Department of Food Studies, Nutrition and Dietetics, Uppsala University; 751 22 Uppsala, Sweden

**Keywords:** built environment, physical activity, diet, behavior, interaction, scoping review, review

## Abstract

A review of reviews following a scoping review study design was conducted in order to deconstruct the black box of interactions between the built environment and human behaviors pertaining to physical activity and/or diet. In the qualitative analysis 107 records were included, 45 of which were also coded. Most review papers confirmed the influence of the built environment on the behaviors of interest with some noting that a same built environment feature could have different behavioral outcomes. The conceptual model developed sheds light on these mixed results and brings out the role of several personal and behavioral factors in the shift from the measured to the perceived built environment. This shift was found to shape individuals’ behaviors critically and to have the power of redefining the strength of every interaction. Apart from its theoretical relevance, this model has high practical relevance especially for the design and implementation of interventions with a behavioral component. Intervention researchers can use the model developed to identify and label the built environment and individual factors that can be measured objectively or perceived as facilitators, concurrent options and barriers, in order to develop comprehensive and multi-component intervention strategies.

## 1. Introduction

Research on the built environment and its relationship to human behaviors has mainly been influenced by urban studies and the development of socioecological models bringing out the complex mechanism of interactions between personal and environmental factors [1]. This approach first developed by urban sociologists turned out to be particularly relevant for the prevention of disease and promotion of health. Research on the impact of the built environment on health-related behaviors has blossomed in the last decades, with an increased focus on physical activity (PA) and diet-related outcomes; both well-known risk factors for many non-communicable diseases [2]. New frameworks were developed such as the Analysis Grid for Environments Linked to Obesity (ANGELO) that advocates for a shift of focus from the individual only, to a broader conceptualization of individuals as interacting with specific economic, physical, sociocultural and political environments [3]. Other related and particularly influential works include the ecological model of active living developed by Sallis et al., 2006 [4] and research on the food environment notably led by Glanz et al., 2007 [5]. Building on this research, several studies have demonstrated that built environment factors such as the availability of transportation systems or the presence of grocery stores providing access to healthy food act as determinants of PA and nutritious dietary practices [6,7,8]. However, these studies have not conceptualized why the availability of such facilities does not necessarily lead to the same behavior in different contexts or even in the same people. In fact, few papers have attempted to deconstruct the black box of interactions which occurs between the built environment and individuals’ behaviors pertaining to PA and/or diet. 

The black box epitomizes the unknown mechanisms at play that connect the built environment with individuals and their behaviors. Deconstructing the key interactions is critical in shedding light on the factors that contribute to the development of obesogenic environments, and in turn, in effectively modifying the built environment and designing better population-wide behavioral interventions. The first aim of this review of reviews was to give an overview of research findings linking built environment features with PA and diet-related behaviors. The second aim was to deconstruct and conceptualize the content of the black box in order to understand the complex mechanisms by which the built environment disrupts individuals’ energy balance and promotes unhealthy behaviors. Hence, this paper does not aim at drawing causality links or systematic associations but at understanding key interaction mechanisms so that the relationship between behaviors and the built environment can be analyzed in different contexts. 

## 2. Materials and Methods

### Study Design

A review of reviews was conducted based on the scoping review methodology developed in 6 stages by Arksey and O’Malley, 2005 [9]. The sixth and optional stage “Consultation exercise”, was not included and we conducted stage 4 “Charting the data” and stage 5 “Collating, summarizing and reporting the results” together as one step. 

Stage 1: Identifying the research questions. The objective of this review of reviews was to develop a conceptual model deconstructing the black box interactions between the built environment, individuals and their behaviors pertaining to PA and/or diet (Figure 1). The aim of such a conceptual model was to better inform interventions encouraging an increase of PA along with better diets. The following research questions were identified: What are the complex mechanisms of interactions through which the built environment is influencing individuals and their behaviors? How do the different factors relate to each other? How do the interactions potentially impact health promotion interventions in this field?

The relevant terms and corresponding definitions applicable for this study are outlined below.

The built environment refers to the human-made features of the physical environment and encompasses all urban or rural buildings, infrastructures, facilities, features and open spaces that are created or modified by humans and in which people live, work and recreate [10,11]. It also includes derived constructs and characteristics relevant to its use such as density or safety [12]. Natural physical environment factors such as the weather, the geology or the air quality were not the focus of this paper. However, they were taken into consideration as needed to deepen the understanding of the interactions at play between individuals and their environment.

Physical activity (PA) was defined by the World Health Organization (WHO) as “any bodily movement produced by skeletal muscles that requires energy expenditure—including activities undertaken while working, playing, carrying out household chores, travelling, and engaging in recreational pursuits” [13]. These actions pertaining to PA were grouped into four domains of active living by Sallis et al., 2006: (1) active recreation; (2) household activities; (3) active transport; and (4) occupational activities [4]. Any self-reported or objectively assessed bodily movement falling under one of these four active living domains was considered relevant to PA for this review.

Diet was defined as the sum of all the food choices made on a daily basis by individuals through a complex interplay of social, cultural, environmental, nutritional and health factors. In this scoping review, the concept of diet was used to refer to both actions of purchasing and/or preparing the foods chosen for consumption. Both objectively measured and perceived aspects of the built environment, PA and diet fell into the scope of this review of reviews including inventories, accelerometers, geographical information systems (GIS) or global positioning systems (GPS). In this paper, PA and diet were addressed in combination because from a behavioral and public health standpoint, they are complementary and often need to be addressed together in interventions.

Stage 2: Identifying relevant review papers. A comprehensive online systematic search of four electronic databases (Cochrane Library, Embase, PubMed, Web of Science) was conducted in February 2018 and updated in March 2019. These databases were chosen with the aim to find a balance between specialist and multidisciplinary sources. The search was limited to review papers published in English during January 2007 to March 2019. This timeframe was adopted because of the significant amount of literature available and recently published on the built environment. The search strategy was first developed and tested in PubMed and later adjusted to the other three online databases (Cochrane Library, Embase and Web of Science). When possible, the search was refined by using the following filters: review, meta-analysis and systematic review. Search terms were identified through the iterative development of keywords lists related to the built environment, PA or diet (Table 1). Search strategies of similar reviews were examined, and the Yale MeSH Analyzer was used to retrieve keywords from a list of 10 relevant selected articles. The final search strategy was reviewed by librarian specialists and specialists in the field of this research.

All four databases were searched with the two following combinations of keywords: (1) built environment AND physical activity; (2) built environment AND diet. This means that in (1) all the key words listed in Table 1 for “Built environment” AND all those listed for “Physical Activity” were used in the online search. The same approach was used for the second combination of keywords pertaining to “Built environment” and “Diet”. For more details, the final search strategy used for each database is outlined in Appendix A.

A manual search of the Active Living Research database was also conducted in March 2018 and updated in March 2019. This multidisciplinary online database administered by the University of California, San Diego School of Medicine, aims at gathering all publications with the latest evidence on policy and environmental changes encouraging more PA [14]. 

Finally, snowball searches were applied to identify additional relevant papers and to search for gray literature such as national and international reports based on systematic reviews [15]. References from included reviews were scanned and potentially relevant papers were screened for acceptance, based on the criteria outlined below. 

Stage 3: Paper selection. Records identified through the online database searches were imported to Mendeley and screened for inclusion. Eligibility criteria were developed before abstract screening including (1) papers published in English between January 2007 and March 2019; (2) papers pertain to human studies; (3) be a systematic review, meta-analysis, scoping review, or any other type of review based on a systematic search strategy; (4) examine some aspect of the built environment in relation to individuals and behaviors pertaining to PA and/or diet; and (5) use perceived, objective or proxy measures of the built environment, PA and diet such as “fast food orders”, "park usage” or any other form of amenity/equipment usage, relevant to PA or diet. Reviews of reviews were selected only if they had included relevant reviews published before 2007. The papers were screened by the first author (AST) and all uncertainties were discussed and assessed by the second (KSA) and last author (MD). Additional information regarding the screening process and respective reasons for articles’ exclusion can be found in Appendix A. A total of 107 review papers were selected after screening and included in the review. 

Stage 4: Collating, charting and summarizing of data. This stage followed three distinct phases: (a) thematic analysis; (b) model development and (c) narrative analysis. Thematic analysis and model development though described separately, were carried out simultaneously.

(a) Thematic analysis. Coding was done in two stages using the thematic analysis approach outlined in Spencer et al. [16] and followed the steps of data familiarization; initial thematic framework development; data indexing and sorting; and data extracts reviewing. Among the 107 papers selected for the review, 30 were first randomly selected for coding, based on their outcome of focus (PA and/or diet): 12 PA, 10 diet, 8 PA and diet. The remaining 77 papers were read in detail and 15 additional papers that differed in terms of topic, target population or outcome focus were identified and included for further coding (Subset 1: 30 + 15 = 45 papers, Appendix A). The unit of qualitative analysis (or one data sample) was the results and discussion sections of each selected paper. Familiarization with the 45 articles facilitated the identification of a primary list of topics and items of interest, that was continually examined across the papers. The inventory of items was then rationalized and structured into a thematic framework in which identified topics and items were hierarchized into descriptive themes and subthemes according to different levels of generality. Topic 1: Behavior; included two descriptive themes: healthy behaviors and unhealthy behaviors. Topic 2: Built environment; regrouped two descriptive themes: built environment dimensions and built environment characteristics. The latter was divided into two descriptive subthemes: measured characteristics and perceived characteristics. All themes and subthemes were grounded in the data retrieved from the coded articles. Themes and subthemes were named, and their meaning was clarified in order to ensure a coherence within the developed thematic framework (Table 2). The thematic framework was then re-applied to the 45 coded papers and data was sorted under the same descriptive themes and subthemes. Data extracts were reviewed, and the thematic framework was refined when new themes or sub-themes were identified. The thematic coding of Subset 1 was carried out in NVivo qualitative data analysis Software; QSR International Pty Ltd. Version 11, 2012. Among the 45 review papers coded, 27 pertained to PA, 10 to diet and 8 to PA and diet.

Charting the data. The following information was systematically extracted and tabulated from the 107 papers included in the review: author(s), year of publication, outcome type (PA, diet, PA and diet), aims of the review paper, target populations, methodology and key findings (Appendix A). This data was charted by the first author and used to depict the characteristics of the review papers in order to contextualize the results of the qualitative analysis. 

(b) Model development. Existing models and frameworks pertaining to behavior and/or the built environment were identified based on their capacity to isolate the different components of these two constructs. The models identified were adapted and used together with the results of the coding to develop an evidence-based model conceptualizing the spectrum of complex interactions occurring between the built environment, individuals and their behaviors pertaining to PA and/or diet.

(c) Narrative analysis. The results from the qualitative analysis were narratively reported. The key findings from subsets 1 and 2 were used to illustrate the logic of the conceptual model developed based on the thematic analysis. The different components of the model were deconstructed and the role they play in the interactions between the built environment and behaviors explained and exemplified. 

## 3. Results

### 3.1. Articles Included in the Review

The electronic search yielded 1000 articles from the four selected online databases (PubMed, *n* = 216; Web of Science, *n* = 464; Cochrane Library, *n* = 37; Embase, *n* = 283). Additionally, 3 reviews were retrieved from the manual search of the Active Living Research database; bringing the total number of records identified through online database searching to 1003. Following the removal of all duplicates, 716 articles were screened for acceptance. After title and abstract screening, 108 full-text articles were assessed for eligibility based on the pre-defined inclusion and exclusion criteria. Of those, 31 were excluded and 77 met the inclusion criteria. A total of 60 additional papers were retrieved from the reference lists which after full-text screening, 30 papers were included and added. As a result, a total of 107 records were included in this scoping review and 45 of those were also coded in Nvivo (Figure 2). 

### 3.2. Characteristics of the Included Articles

Details on the selected characteristics of the articles included in this scoping review are outlined in Table 3 and in Appendix A. A total of 83 (78%) were focusing on the relationship between the built environment and PA whereas 13 (12%) of them examined diet outcomes. Approximately the same share of articles (10%) were examining both PA and diet outcomes in relationship to the built environment. Most of the records were either systematic reviews (62%)—with very few performing a meta-analysis—or reviews based on a systematic search (30%). The articles examined diverse environment settings and focused on different population groups and subgroups such as children, adults, older adults, women, ethnic minorities, persons with disability or low-income populations. The range of included studies in the reviews varied from 5 [35,60] to 440 [61]. 

### 3.3. Description of the Conceptual Model

Based on the thematic and narrative analyses, seven key components were identified that are essential to conceptualize the complex mechanism of interactions at play between the built environment, individuals and their behaviors pertaining to PA and/or diet. Figure 3 depicts the key components numbered 1–7 and their inter-relationships using broken or unbroken lines named a–k. These numbers and alphabets are referred in the text within brackets (). The model draws on two existing models, one on behavior and the other on built environment to shed light on the black box (Figure 1) depicting interactions between the two [62,63]. The Capability, Opportunity, Motivation and Behavior (COM-B) model developed by Michie et al., 2011, identifies four sources of behavior relevant for intervention (capability, opportunity, motivation and behavior) and places them at the core of an interaction system. Turner et al., 2018, categorize the food environment characteristics into either an external or an internal domain. Individuals’ perceptions and experiences reshape the characteristics of their environment and lead to a shift from the first to the second domain. The results of the thematic and narrative analyses are used to demonstrate how the relationship between capability, opportunity, motivation and behavior is being impacted by the interactions of these four components with the internal and external dimensions of the built environment. 



**1**

**PERSONAL FACTORS**
The starting point for the model could be personal factors which are divided into two categories. The first one is intrinsic to individuals and encompasses factors such as cognition and affect (intention, attitude, preferences, emotions) as well as physical characteristics (health condition, age, sex). The second one is extrinsic and covers social aspects such as socioeconomic status and cultural background. 
**2**

**EXTERNAL BUILT ENVIRONMENT**
The external built environment is defined by five characteristics that are exogenous to individuals: availability; security and safety; price; marketing and aesthetics; and the properties of vendor, product, infrastructure or facility. 
**3**

**CAPABILITY**
Individuals’ capabilities are divided into four categories: biological, psychosocial, educational (skills/knowledge) and economic.
**4**

**INTERNAL BUILT ENVIRONMENT**
The way individuals experience the external built environment creates a new subjective lens that leads to a shift from the external (objective) to the internal dimension of the built environment (perceived). 
**5**

**OPPORTUNITY**
Opportunities are the addition of all the objective and perceived barriers, facilitators and concurrent options emerging from the internal and the external built environment.
**6**

**MOTIVATION**
As described by the Capability, Opportunity, Motivation and Behavior (COM-B) framework, motivation [6] is also central to this conceptual model. Here, motivation is being deconstructed into two main dimensions: decision and action.
**7**

**PA and DIET BEHAVIOR**
PA and diet-related behaviors are the main outcomes of this model. They are represented by a spectrum, ranging from the unhealthiest (sedentariness, unhealthy diet) to the healthiest behaviors (vigorous PA, healthy diet). 
**a**
Personal factors (1) are related to the external built environment (2) through a mechanism of neighborhood self-selection whereby personal preferences pertaining to PA and diet could be reflected in individuals’ choice of neighborhood. It is an interrupted line because the process of self-selection is not systematic and is influenced by economic, social and work-related constraints. 
**b**
Personal factors (1), such as cognition and physical characteristics, shape individuals’ capabilities (3). Educational, economic and psychosocial capabilities are impacted by social determinants and other factors such as age and attitude.
**c**
The external built environment (2), such as the availability of adapted facilities tailored to their special needs, shapes individuals’ capabilities (3). Educational and economic capabilities are impacted by the availability or price of urban and food facilities’ characteristics. Psychosocial capabilities are influenced by factors such as the security/safety and aesthetics of the built environment. 
**d**
Depending on their own capabilities (3), individuals evaluate specific built environment features as more or less accessible; of greater or lesser safety; as more or less affordable; or as more or less convenient (4).
**e**
The shift from the external (2) to the internal built environment dimension (4) is epitomized by the emergence of perceived facilitators, barriers and concurrent options (5). These reshape individuals’ opportunities and can in turn increase or decrease their motivation to achieve healthy behaviors.
**f**
Opportunities (5) for PA and healthy diets are initially defined by the external built environment characteristics (2) and the resulting facilitators, barriers and concurrent options, which provide objectively measured opportunities for individuals to be physically active or to adopt and maintain healthy diets.
**g**
Motivation (6) is shaped by all the above-mentioned personal factors (1).
**h**
Motivation (6) influences individuals’ perception of opportunities (5) for PA and healthy diet.
**i**
Motivation (6) modifies individuals’ perceptions of their own capabilities (3) (self-efficacy).
**j**
Motivation is also impacted by the shift from the objective to the perceived built environment, which leads to weaken or strengthen individuals’ decision to perform a healthy behavior.
**k**
This model does not only conceptualize behaviors (7) as outcomes but also as additional key factors at play in the interaction between the built environment, individuals and future behaviors. In fact, previous PA and diet behaviors (7) influence future behaviors by modifying personal factors (1) such as individuals’ preferences and habits.


### 3.4. Description of Key Components

The key components of the conceptual model (1–7) are further described and exemplified here:

The built environment. Multiple built environment features were retrieved during the coding phase with some reviews examining a significant set of built environment attributes [8,51,53]. These different features, factors and attributes were classified into two domains based on the food environment framework developed by Turner et al., 2018 [63]. These two domains are: the external built environment (2) and the internal built environment (4) as shown in Figure 3. The first one is exogenous and refers to all the features of the built environment that can be objectively measured such as the overall number of destinations in a neighborhood [53], the total number of crimes recorded [54], or the type of labelling used on foods such as vegetables [46]. The second one is endogenous and refers to the perception of those same characteristics by individuals, in terms of their needs and capabilities. This distinction between the external and the internal built environment is essential as behavioral outcomes cannot often be explained by measured external components alone [63]. For example, McCormack and Shiell, 2011, found in their review that individuals who expressed a preference for less walkable neighborhoods are generally less likely to walk whether the built environment in which they lived is supportive of walkability or not [48]. This demonstrates that the availability of built environment features supportive of walking and the outcome behavior of walking, are not directly connected. In fact, the relationship described is moderated by personal factors such as preferences as well as by other factors pertaining to motivation. The same finding was supported by Cerin et al., 2017, who found that the presence of stray animals is modifying the observed association between public transport availability and walking [20]. Therefore, the five aforementioned characteristics of the internal built environment, complement the five characteristics of the external built environment. 

Opportunity. The proposed model depicts opportunities as a set of facilitators, barriers and concurrent options that are defined both by the external and the internal built environment. For example, the opportunity to walk to a supermarket within walkable distance from home depends on each individual’s capability to walk without hardship [54]. Facilitators can then be perceived as barriers. Eisenberg et al., 2017, found that wheelchair users were assessing barriers to accessibility not only in terms of the presence/absence of certain physical features such as steps, or slides; but also in terms of their properties such as their slope inclination [54]. Additionally, several reviews pointed out the role of concurrent options in individuals’ decision-making process [36,37,40,44]. Correa et al., 2015, evoked the “dual role of supermarkets” that provide a wide range of both healthy and unhealthy choices [44]. Krolner et al., 2011, also pointed out in their review the access to unhealthy foods as a factor influencing individuals’ choices even when they were in presence of healthy food choices [36]. The same findings led Larson et al., 2009, to advocate for an improved access to supermarket while decreasing in parallel the access to convenience stores [37]. Regarding PA, several reviews found that car ownership influenced adults’ decision not to use an active mode of transportation [39,64].

Motivation. Motivation is the factor at the core of this model that is converting intentions, capabilities and opportunities into behaviors. The reviews outlined its relationship with personal factors and the key role it plays in shaping individuals’ perceptions of their capability. Yen et al., 2014, found that a personal factor such as age provokes a change in the self-efficacy threshold of individuals. This shift is particularly significant in older adults whose motivation for PA and self-efficacy tend to decrease with age [50]. In that sense, motivation impacts individuals’ perception of their own capability and modifies in turn their perception of the built environment. However, the built environment also impacts motivation in return depending on the opportunities it offers to perform a healthy or an unhealthy behavior related to PA or diet. For example, Yen et al., 2014, noted that aspects of the built environment enhancing the feeling of safety such as aesthetics or the absence of litter are shaping opportunities for PA and positively influence motivation and self-efficacy [50]. 

Behaviors. Regarding diet, the main outcome behaviors examined by the reviews were the purchase and consumption of certain types of foods [8,21,22,24,33,34,35,36,37,38,39,40,41,42,43,44,45,46,47]. PA-related behaviors varied between measures of total PA [25,51,53], PA levels (light, moderate, vigorous) [22,24,25,27,30,48,51,57], walking [12,25,26], biking [24,28,29] and exercising [30,31,32]. Outcome behaviors are essential in this model (Figure 3) because previous PA and diet behaviors were found to be correlated to future PA and diet behaviors [29]. Additionally, Pucher et al., 2010, refer to “safety in numbers” to explain that, as more individuals cycled, the safer they feel and the more they tend to use their bike in return [29]. In that sense, previous behavior and others’ behavior both feed back into the model that was developed (Figure 3) and directly impact personal factors such as preferences. This mechanism creates habits and automatisms leading behaviors to be less dependent on the supportiveness of the built environment than on individuals’ inner motivation to pursue their desire. Hence, not only are individuals’ behaviors influenced by their own previous behaviors, but also by others’ behaviors. However, previous behavior and preferences do not only impact motivation but also the built environment through the mechanism of self-selection [48]. The latter was a particular point of focus in the review by McCormack et al., 2011, and occurs each time that health conscious residents with a preference for PA and/or healthy diets choose to live in a particular locality that will increase their opportunities for physical activity and/or healthy foods. 

## 4. Discussion

Based on 107 review articles, a conceptual model was constructed with the aim of depicting the black box of interactions between the built environment and behaviors related to PA and/or diet. The key results of the narrative analysis confirmed the link between the built environment and the behaviors of interest. Most review papers found a positive association between accessibility and healthy diet or between the availability of sport facilities or transport infrastructures and PA. However, mixed results were also found across the different reviews for a same built environment attribute. This can be explained by the emphasis placed by some primary studies or review papers on a single built environment feature instead of a group of features [29]. In their review Pucher et al., 2010, suggested that it was often too complex and not always relevant to attribute specific PA-related behaviors such as biking to specific built environment features such as the availability of bikes. Instead, policy makers and intervention research should attempt to implement programs which simultaneously improve the quality of bicycling infrastructures and traffic security, multiply the number of bike lanes and parking while increasing the availability of bikes. Such approaches were successfully adopted by cities like Bogota or Paris which developed comprehensive citybike programs and saw the total number of bike travels significantly increase over the past few years [29]. Additionally, papers focusing on a single built environment feature miss out on the complex interrelation between the different dimensions of the built environment [65]. In fact, focusing on the availability of supermarkets without looking at the concurrent proximity of convenient stores does not allow a proper understanding of the relationship between supermarket access and healthy diet [37]. Lee et al., 2016, found that most “interventions were ineffective unless fundamental issues were addressed” such as issues pertaining to cost and availability. This means that barriers to PA need to be addressed simultaneously in order to have a real impact on individuals’ behaviors [60]. These findings must lead intervention researchers to ask fundamental questions such as: what is the sine qua non condition without which the relationship between a specific built environment feature and an outcome behavior cannot happen? In other words, despite all facilitators, is there a built environment feature constituting a barrier that cancels the effect of other facilitators for a specific target population? 

By deconstructing the mechanism of interactions, the conceptual model attempts to give a more complete understanding of the gaps in existing models and the mixed results found in the literature. Relevant existing models are essentially of two types: the socioecological models with a major focus on the environment and the behavioral models focusing on the individual. The socioecological models were comprehensive to the extent that they considered all levels of environmental influences. However, they did not allow a deconstruction of the interactions of interest [4,66,67]. For example, the Social-Ecological Model for Food and Physical Activity Decisions developed by the Centers for Disease Control and Prevention, USA, conveyed a good understanding of how the four identified ‘layers of influence’ (social and cultural norms and values, sectors, setting, individual factors) shape both individuals’ food and beverage intake as well as PA levels. However, the model does not elucidate the interactive dynamics linking those intersecting ‘layers of influence’ and their relationship to the behaviors in question [66]. Behavioral models, on the other hand, tend to focus on one type of outcome such as mobility in older adults without taking a closer look at the overall interaction between behavioral and built environmental factors [50]. In their model on the food environment, Turner et al., 2018, suggest a middle-ground by conceptualizing a binary complementarity between one external and one internal built environment characteristics: availability-accessibility, security and safety-the feeling of security and safety etc. [63]. The model developed in this study (Figure 3) takes it a step further and sees larger connections between safety and security, aesthetics, facility properties and the feeling of security and safety. This suggests that several external built environment dimensions contribute to define one internal built environment dimension and that a measured built environment characteristic such as aesthetics (presence of litter, pleasant scenery)—a priori unrelated to the perceived built environment characteristic of security and safety feeling—can trigger a feeling of security and/or safety [50]. 

In that sense, the main innovation proposed by the conceptual model (Figure 3) was to shed light on the dynamic relationship between the measured and the perceived built environment. In this model, the two domains of the built environment are simultaneously considered because of their concurrent role in shaping capabilities, opportunities, motivation and eventually behaviors. This model does not aim at drawing causality links or systematic associations but at understanding key mechanisms so that the relationship between behaviors and the built environment can be analyzed in different contexts. Therefore, the role of context in shaping these interactions also becomes more explicit in this model. Furthermore, the model integrates a two-way dynamic between the built environment and behavior that is not part of existing models such as the socioecological model. In the latter, the environment was considered as the highest level of influence and individual factors were depicted as the manifestation of these influences in individuals. This conceptual model adds another dynamic according to which not only the built environment supports behaviors, but behaviors also shape the built environment.

### 4.1. Key Implications of the Conceptual Model for Interventions

The key implication of this conceptual model is that behavioral interventions with a built environment focus should: (1) identify the specific built environment features at play in a specific context during the development phase; (2) index them as facilitators, barriers or concurrent options depending on the target population (meaning that for different target populations, a same built environment feature can be categorized as a facilitator, barrier or a concurrent option); (3) define whether the built environment feature identified as a barrier or concurrent option is a fundamental issue or not (i.e., a feature whose absence or presence will cancel the effect of a built environment facilitator); (4) if it is a fundamental issue, concentrate the resources on it; if not, focus the resources on reinforcing facilitators and ideally, strengthen facilitators while reducing concurrent options and suppressing barriers. Moreover, context is a key factor in shaping interventions as the relationship between the measured and perceived built environment is highly context-dependent. This is relevant both for the development and implementation of any interventions with a behavioral component [68]. 

### 4.2. Methodological Considerations

An exploratory search was run prior to initiating this scoping review and revealed the large available body of review papers covering one or several aspects of our research question. A scoping review of reviews appeared to be the most relevant methodology to gain a comprehensive understanding of the interactions of interest. However, findings from primary studies that were not captured by any of the review papers screened are potentially missing from this review of reviews. This could notably include findings from the most recent literature that has not been included yet in any review paper or findings covering specific aspects of the built environment that have never been captured by any of the review papers published so far.

The principal strength of this review of reviews, nonetheless, lies in the rigorous and transparent search strategy applied for the literature screening. This strategy was ideal to retrieve a large number of review papers and develop a conceptual framework grounded in the existing literature. Review papers were screened and coded by a single reviewer and cross-checked by a second reviewer. A third reviewer arbitrated when uncertainties were raised by the first and second reviewers. The evidence coded during the qualitative analysis was not weighted according to the occurrence of themes, nor the quality of papers appraised, because the principal objective of this scoping review was to understand the mechanism of the interactions at play by deconstructing them. As a result, findings mentioned only by few reviews were considered as relevant if they were highlighting a new dimension of an interaction between the built environment and behaviors. 

Referring to the mixed results alluded to in the discussion, the present study is limited by the studies included in the reviews. Several reviews highlighted the limitation of the cross-sectional study design often used in the primary studies [8,43,52]. In fact, cross-sectional studies do not allow any conclusion about causality to be drawn. The significant gaps observed in the measurement methodologies adopted by review papers were also pointed out by Ding et al., 2012, as an important issue limiting opportunities for comparisons between studies [69]. Furthermore, there was a high inconsistency in the definitions used across review papers to characterize the built environment, either objectively measured or perceived. These discrepancies were acknowledged, and a thematic framework was developed in order to index and sort the data in a systematic way, thereby enabling comparisons across studies and reviews. 

Most of the methodological concerns are intrinsic to the scoping review design; however, the choices made were considered as necessary to allow the framework to be applied in different contexts. The adaptability of the model makes it dynamic and allows for high context specificity, which will ultimately help public health actors to understand the role of personal and built environment factors in relation to each other and develop high-impact approaches for interventions. However, it is important to note that the model was developed predominantly from evidence drawn from studies in high-income settings, and only limited evidence from low-and middle-income countries [58,60,70].

## 5. Conclusions

The conceptual model depicts the black box of interactions and deconstructs the complex relationship between measured and perceived aspects of the built environment as well as the interactive role of behavioral and personal factors. The conceptualization of the shift from the external to the internal built environment is critical in understanding the environmental and individual dynamics leading to healthy and unhealthy behaviors related to PA and/or diet. Apart from its theoretical relevance, this model has high practical relevance especially for the design and implementation of interventions with a behavioral component. This paper provides a methodological tool that will help academics, policy makers and the public to deconstruct the interactions and help them better understand the way individuals interact with their built environment. For example, this model would be of particular interest in the first analytical phases of urban projects aimed at making cities more accessible to non-motorized means of transportations or aimed at encouraging the development of food stores selling local products. Intervention researchers can also use the model to identify and label the built environment and individual factors that can be objectively measured or perceived as facilitators, concurrent options and barriers in order to develop comprehensive and multi-component intervention strategies.

## Figures and Tables

**Figure 1 ijerph-16-01454-f001:**
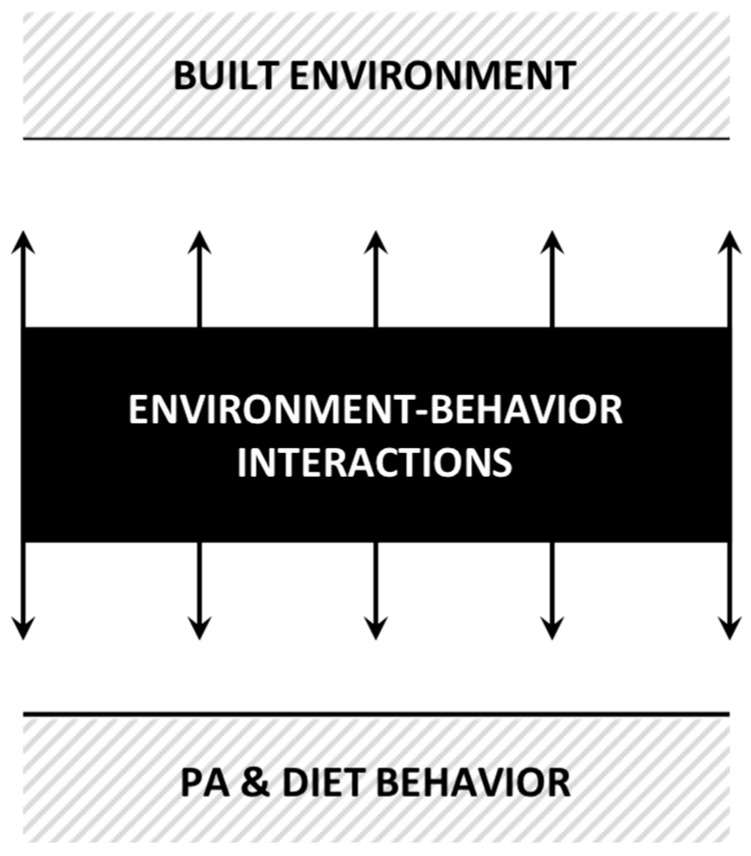
Black box of interactions: the focus of the scoping review.

**Figure 2 ijerph-16-01454-f002:**
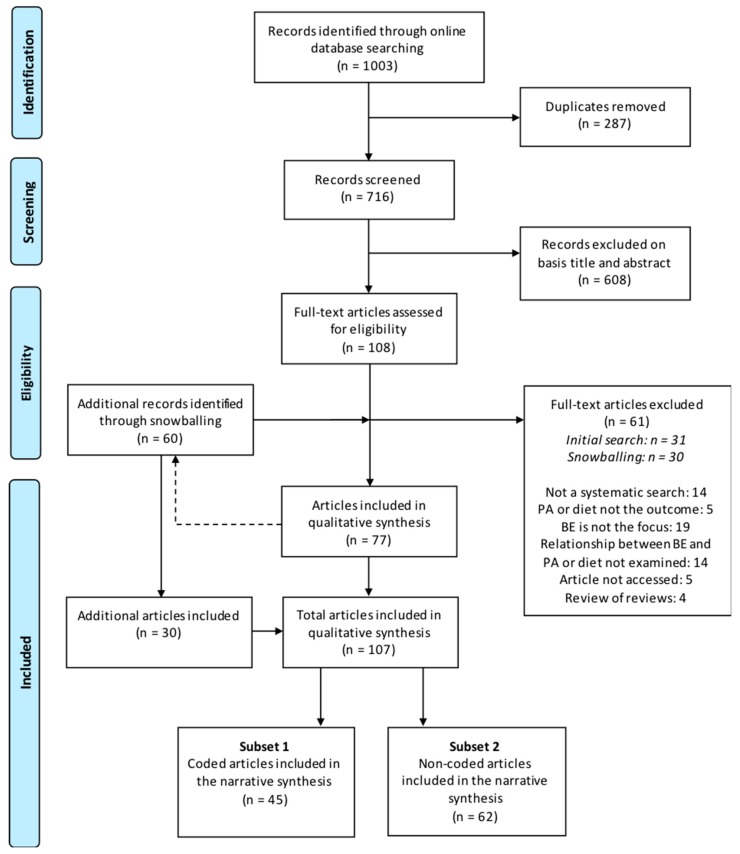
PRISMA flowchart of identification and selection of review papers for inclusion in the scoping review. Note: Figure adapted from Liberati et al., 2009 [59].

**Figure 3 ijerph-16-01454-f003:**
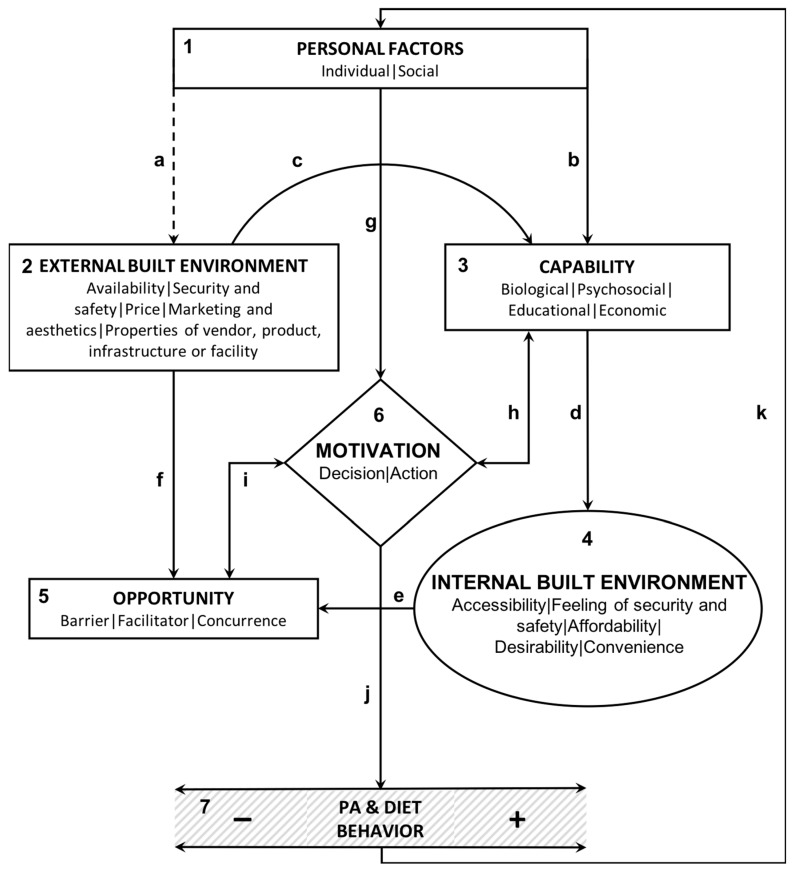
Conceptual model depicting the interactions between the built environment, individuals and their behaviors.

**Table 1 ijerph-16-01454-t001:** Keywords used in the search strategy.

Construct	Key Words
Built Environment	Built environment, physical environment, neighborhood environment, urban form, urban planning, city planning, environmental planning, environment design, urban design, architecture, streetscape, foodscape, spatial behavior
Physical Activity	Physical activity, physical inactivity, inactivity, sedentary lifestyle, sedentary life style, sedentary behavior, sedentary behaviors, exercise, sport, sports, running, walking, walk *, strolling, biking, bike, bicycle, bicycling, cycling, commuting, active commuting, active transportation, active travel, commuting, transportation modes, transportation choices, school recess, active recreation, active living
Diet	Diet, food, nutrition, healthy diet, unhealthy diet, unhealthy diets, unhealthy food, unhealthy foods, feeding behavior, eating, eat, food choice, food choices, food decisions

Note: Truncation is represented by an asterisk (*). The use of truncation allows to retrieve various word endings and spellings. Example: walk *—finds also walking, walks, walkability etc.

**Table 2 ijerph-16-01454-t002:** Thematic framework used for coding subset 1.

Descriptive Themes and Sub-Themes by Topic	Elements of Descriptive Themes and Sub-Themes	Definition and Example
Topic 1: Behavior		
1.1 Descriptive theme: Healthy behavior		
• Physical activity	Active commuting, recreational, household, occupational, exercise	Physical activity outcomes were divided into 4 active living categories based on the framework developed by Sallis et al., 2006: (1) active recreation; (2) household activities; (3) active transport; and (4) occupational activities [4]. The main PA outcomes examined in the review papers were: active travel [17,18,19,20,21,22,23,24], recreational walking [12,25,26], active recess [27], biking [24,28,29], exercising [30,31,32]
• Healthy diet	Purchase, intake, consumption of healthy foods (fruit and vegetables, low fat, high fiber, non-meat protein, whole grain)	Healthy diet was defined as both the consumption of healthy foods in a sufficient quantity and the non-consumption of unhealthy foods. Fruit and vegetable (FV) intake as well as low fat consumption were the most examined healthy diet outcome behaviors [8,21,22,24,33,34,35,36,37,38,39,40,41,42,43,44,45,46,47]
1.2 Descriptive theme: Unhealthy behavior		
• Physical inactivity	Sedentary time, sitting time, low physical activity	Sedentary behaviors were defined as the quasi absence of bodily movements as opposed to PA and ranged from TV watching, video-gaming in the home environment to sitting in the workplace environment [21,30,38]
• Unhealthy diet	Purchase, intake, consumption of unhealthy foods (high/ saturated fat, sugar-sweetened, red meat)	Unhealthy diet was defined as the purchase and consumption of foods leading to unhealthy weight gains and increased risks of chronic diseases such as diabetes. Behaviors such as high fat consumption, fast foods consumption or low FV intake were examined by several reviews and were indexed as unhealthy diets in this thematic framework [8,21,22,24,33,34,35,36,37,38,39,40,41,42,43,44,45,46,47]
Topic 2: Built environment		
2.1 Descriptive theme: Built environment dimensions		
• Design	Street network, transport infrastructure, open space, building infrastructures (playground markings, stairwells) neighborhood/home/work/school/ recreation/service facilities and space, detached or fixed facilities and equipment, service facilities (restaurants or grocery stores)	Design was defined as the category grouping all the characteristics and aspects of the physical human made structures created and used by humans for a purpose. The physical structure of roads, sidewalks, stairs, playgrounds or restaurants were often analyzed in the review papers [21,22,24,25,32,35,48,49]
• Destination	Number, type (transport related, grocery related, work related, neighborhood related)	Destination referred to all physical indoor or outdoor spaces offering some type of service or goods where individuals would go with a purpose such as: shops, churches, schools, workplaces, lakes etc. [25,48,50] To that extent, destination is a more abstract construct than design. Sugiyama et al., 2012 made destination a central theme in their review [26]
• Land use	Land use repartition; spatial repartition of building, street networks, infrastructures, and facilities; sprawl; urbanization; residential vs. non-residential; urban vs. rural	Land use referred to the spatial repartition of infrastructures, facilities, buildings and destinations. This dimension was often examined in terms of mix, diversity and accessibility of destinations such as shops, services or food shops [12,18,19,20,22,25,26,28,48,49,50,51,52]
2.2. Descriptive theme: Built environment characteristics		
2.2.1 Descriptive sub-theme: Measured characteristics (external/exogenous)		
• Availability	Presence, number, ratio, density, diversity	Availability was defined as the quantitative evaluation of the presence of an infrastructure/facility, a destination or foods available for use or consumption. This characteristic was mainly characterized in the review papers in terms of destination availability and equipment or facility availability [28,34,41,51,53]
• Security and safety	Traffic safety, personal safety, safety from crime, infrastructure security/safety	Security and safety were defined as involving both interactions with other individuals (drivers, criminals etc.) and infrastructures/facilities concerns (security/safety of roads, public transport etc.). This characteristic was mainly examined in terms of personal safety, notably from a pedestrian point of view (crime and personal related security/safety) [42,53], but also in terms of traffic injury for both pedestrians and cyclists [17,51].
• Price	Monetary value of a food product, a service, a facility, an infrastructure	Price was defined as the monetary value of foods and facility use. This characteristic was examined in several review papers, notably those focusing on food price [43,44,45]
• Marketing and aesthetics	Product marketing and branding (visibility, placement, promotion, nudges), built environment aesthetics, natural elements (greenery and pleasant scenery), absence of liter and signs of disorder	This characteristic referred to all the strategies used to present built environment features or foods placed in built environment structures such as grocery stores as appealing and visually pleasant. This external built environment characteristic was found to influence food consumption or facility/equipment use by several reviews [26,33,54,55]
• Property of vendor, product, infrastructure or facility	Product properties (quality, color), type of vendor (restaurant, fast foods, grocery store, store opening hours), infrastructure/facility properties (quality, length, height)	This characteristic referred to all physical or non-physical features that defined the functioning of an object, a place or a food as well as its intrinsic characteristics. For example, Kaushal et al., 2014 found that the size of sport equipment available at home was influencing individuals’ PA levels [30]. Larson, 2009 found that product properties such as food quality were often associated with certain types of vendors [37]
2.2.2 Descriptive sub-theme: Perceived characteristics (internal/endogenous)		
• Accessibility	Proximity, distance, time, perceived diversity	Accessibility referred to each individual’s capability to reach the diverse destinations, goods or facilities available for use or consumption. This was a central theme in many reviews [17,21,26,31,32,37,44,47,49,56,57]
• Feeling of safety and security	Perceived traffic safety, perceived personal safety, perceived safety from crime, perceived infrastructure safety	This characteristic was defined by individuals’ perception of their environment security and safety. Crime rates and traffic injuries were not necessarily related to people’s feeling of security and safety. Other external built environment features contribute to people’s lack of security/safety feeling such as the absence of lightning [42,58]. For example, one review focusing on older adults’ mobility found that a personal factor such as age was influencing individuals’ perceptions of their built environment and its safety [50]
• Affordability	Individuals’ economic power, socioeconomic status (SES)	This characteristic referred to individuals’ economic capability to pay for a service or products such as public transports or foods. Affordability was notably examined by Engler-Stringer et al., 2014 in their review on the influence of the community and consumer nutrition environment on children’s diet [45]
• Desirability	Perceived built environment aesthetics, natural elements (greenery and pleasant scenery)	Desirability was conceptualized as people’s capability to appreciate the pleasant features of the built environment such as aesthetics and green scenery but also to appreciate the way food products are presented (i.e.: do they consider them as appealing or not?) This was outlined by Cerin et al., 2017 who found that there was a stronger association of PA with greenery in individuals without visual impairment [20]. Healthy foods desirability was also examined by Kahn-Marshall and Gallant, 2012 who tried to understand how individuals’ preferences, taste and knowledge could be influenced in order to *“make healthy behaviors the easy choice for employees”* [34].
• Convenience	Relative time and effort to eat a product or to use a facility/ infrastructure (restaurant, grocery store, shop, public transport, city bikes)	Convenience referred to individual’s capability to proceed with buying or eating foods or using transport facilities and being physically active without difficulty. For example, Krolner et al., 2011 examined how eating settings and food visibility influenced children’s fruit and vegetable consumption [36].

**Table 3 ijerph-16-01454-t003:** Selected characteristics of the review papers included in the qualitative analysis (*n* = 107).

Reviews	*n* (%)
Outcome type	
Physical Activity (PA)	83 (78%)
Diet	13 (12%)
PA and Diet	11 (10%)
Environment setting	
Neighborhood (street, outdoor and public space, transport infrastructure, restaurants, grocery stores etc.)	93 (87%)
Home	2 (2%)
School	7 (7%)
Work	3 (3%)
Multiple settings	2 (2%)
Target population	
General population	29 (27%)
Children/Adolescents	36 (34%)
Adults	22 (21%)
Older adults	7 (7%)
Multiple population groups	13 (12%)
Disadvantaged population group	
Person with disability	2 (2%)
Socially disadvantaged (low Socio-economic status, minorities)	4 (4%)
Study design	
Systematic review	66 (62%)
Meta-analysis	1 (1%)
Systematic review & meta-analysis	5 (5%)
Reviews *	32 (30%)
Review of review	3 (3%)
Number of included studies	
<20	23 (22%)
20–50	61 (57%)
>50	23 (22%)
Publication year	
2007–2013	51 (48%)
2013–2019	56 (52%)

* Reviews based on a systematic search that are not classified as systematic reviews.

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
