# Peer review of "Built Environment and Health Behaviors: Deconstructing the Black Box of Interactions—A Review of Reviews"

_ijerph, 2019, doi:10.3390/ijerph16081454_

Round 1

Reviewer 1 Report

Thank you for the opportunity to review this paper. I read the title with great interest but I would like to acknowledge that I am not an expert in qualitative study. Therefore, some of the comments may or may not be relevant.

The aim of this paper was to deconstruct the ‘black box’ of the relationship between the built environment and physical activity and/or diet. A conceptual model was developed based on thematic and narrative analysis that are supposed to answer questions like: “what are the mechanisms linking the availability of active travelling facilities such as bike lanes to specific behaviours such as biking to school; and why does such facilities not necessarily lead to the same behaviour in different people? What are the paths through which the built environment is influencing food consumption? What are the interaction mechanisms common to both PA and diet related behaviours?”. However, I am not sure if the author has sufficiently addressed the aims by answering these questions in the study. The specific built environment features (bike lanes, recreation facilities, trails etc) on physical activity was not examined, but rather the overall availability of the built environment was included. Due to the lack of specificity in deconstructing the relationship between the built environment and behaviour, I believe the mystery of this black box remains.

I have a few major concerns for this study and most of the concerns are on the development of themes and conceptual framework.

1.       Is it a common practice to conduct qualitative study from quantitative findings? Usually, a qualitative study is conducted to inform quantitative study, but this study seems to be doing the opposite.

2.       Looking at the list of included review studies, there are a lot of variation in the relationships between the built environment features and behaviour (PA/diet). I was initially expected to see how this study would deconstruct the black box of different built environment features (recreation facilities, parks, trails etc) on behaviour but based on the conceptual model (fig3), built environment features are clumped together as ‘external built environment’. How do you develop theme/code in Nvivo if one of the built environment of interest was not associated with physical activity?

3.       How does the author assign each arrow in the mode, why are some bi-directional arrow and some unidirectional? Was it based on the findings of the included studies? The author stated that ‘…interrupted line because the process of self-selection is not systematic and is influenced by economic, social and work-related constraints.’ However, I am still not sure if I understand the meaning of interrupted lines. I believe what was stated in the interrupted line may also be applicable to some other lines in the conceptual model not just from (1) to (2). For example, (3) capability of individual can self-select themselves into (2) external built environment and (5) opportunity. However, if these arrows are assigned based on the findings from the included articles, I would like to see a clearer explanation somewhere (in text).

4.       Are the sub categories (e.g. individual and social from (1) personal factors) stem from the included studies?  There seems to be a lot of overlap between components of the sub-categories. For example, there are overlap between personal factors and capability; as well as external built environment and opportunity.  To give a more specific example, (1) personal factors which included ‘social-socioeconomic status’ as the sub-category can potentially overlap with (3) capability that included ‘educational, economic’. I would like to know how the author distinguish between the two.

5.       I believe the author’s statement about socioecological model is inaccurate. A core principle of ecological models is that influence from multiple levels can interact and exert synergistic effects on behaviour. Papers below justified my statement.

Giles-Corti B, Timperio A, Bull F, Pikora T. Understanding physical activity environmental correlates: Increased specificity for ecological models. Exerc Sport Sci Rev. 2005;33:175-181.

Hovell MF, Wahlgren DR, Gehrman C. The behavioral ecological model: Integrating public health and behavioral science. In: DiClemente RJ, Crosby R, Kegler M, eds. New and emerging theories in health promotion practice & research. San Francisco: Jossey-Bass Inc; 2002.

Ding, D., Sallis, J.F., Conway, T.L., Saelens, B.E., Frank, L.D., Cain, K.L. and Slymen, D.J., 2012. Interactive effects of built environment and psychosocial attributes on physical activity: a test of ecological models. Annals of Behavioral Medicine, 44(3), pp.365-374.

6.       Regarding the systematic search of articles, why didn’t the author include articles published in 2018? There are a few reviews relevant to built environment and PA/diet during 2018-2019. For example,

Stappers, N.E.H., Van Kann, D.H.H., Ettema, D., De Vries, N.K. and Kremers, S.P.J., 2018. The effect of infrastructural changes in the built environment on physical activity, active transportation and sedentary behavior–A systematic review. Health & place, 53, pp.135-149.

MacMillan, F., George, E., Feng, X., Merom, D., Bennie, A., Cook, A., Sanders, T., Dwyer, G., Pang, B., Guagliano, J. and Kolt, G., 2018. Do natural experiments of changes in neighborhood built environment impact physical activity and diet? A systematic review. International journal of environmental research and public health, 15(2), p.217.

Ding, D., Nguyen, B., Learnihan, V., Bauman, A.E., Davey, R., Jalaludin, B. and Gebel, K., 2018. Moving to an active lifestyle? A systematic review of the effects of residential relocation on walking, physical activity and travel behaviour. Br J Sports Med, 52(12), pp.789-799.

Rhodes, R.E., Saelens B.E., Sauvage-Mar, C., 2018. Understanding physical activity through interactions between the built environment and social cognition: a systematic review. Sports Med, 48(8), pp. 1893-1912.

Author Response

Response to Reviewer 1 Comments

Remarks:

-       We updated the review and ran the same search strategies in the same databases in order to include all relevant papers published between 2017 and 2019.

-       Changes in the supplementary files and tables were highlighted in yellow.

-       Changes in the manuscript can be seen in track change and were copied in bold and darker red when integrated to the responses bellow.

General comment:  The aim of this paper was to deconstruct the ‘black box’ of the relationship between the built environment and physical activity and/or diet. A conceptual model was developed based on thematic and narrative analysis that are supposed to answer questions like: “what are the mechanisms linking the availability of active travelling facilities such as bike lanes to specific behaviours such as biking to school; and why does such facilities not necessarily lead to the same behaviour in different people? What are the paths through which the built environment is influencing food consumption? What are the interaction mechanisms common to both PA and diet related behaviours?”. However, I am not sure if the author has sufficiently addressed the aims by answering these questions in the study. The specific built environment features (bike lanes, recreation facilities, trails etc) on physical activity was not examined, but rather the overall availability of the built environment was included. Due to the lack of specificity in deconstructing the relationship between the built environment and behaviour, I believe the mystery of this black box remains.

Response to general comment: The introduction section has been revised and the aim of the study clarified [lines 28-58]. The changes made specify that the aim of this scoping review was not to examine causal relationships but rather to bring out the complexity of the interactions’ pathways. In order to do so, we tried to deconstruct the black box of interactions by identifying interactions’ pathways linking the 7 following dimensions: personal factors, external built environment, capability, internal built environment, opportunity, motivation and behavior. These dimensions and the way they interplay to create interactions between the built environment and PA and/or diet, are grounded in the evidence that emerged from the systematic literature search. Hence, this paper aimed at conceptualizing these interactions ‘pathways and future research should try to validate the model by testing it on interactions between specific built environment features and specific health related behaviors.

In that sense, the black box is an analogy of what we are looking for and Figure 3 (the conceptual model) describes what we found behind the black box.

We addressed the reviewer’s comment by revising the introduction in order to make the aim of the black box analogy clearer but decided to keep figure 1 based on other reviewers’ comments.

Point 1: Is it a common practice to conduct qualitative study from quantitative findings? Usually, a qualitative study is conducted to inform quantitative study, but this study seems to be doing the opposite.

Response 1: As described by Creswell et al (2003), there are different approaches to using qualitative and quantitative methodologies (i.e. sequential explanatory design, sequential exploratory design, sequential transformative design). Further, there are different types of review papers. This review of reviews is synthetizing qualitative and quantitative research with the aim to gather all relevant evidence that will help to deconstruct the interactions’ mechanisms between the built environment and behaviors pertaining to PA and/or diet.  In order to do so, this review of reviews was conducted based on the scoping review methodology developed in 6 stages by Arksey & O’Malley, 2005 (Arksey H, O’Malley L. Scoping studies: Towards a methodological framework. Int J Soc Res Methodol Theory Pract. 2005;8:19–32). In their methodology, Arksey & O’Malley notably claim that “rather than being guided by a highly focused research question that lends itself to searching for particular study designs (as might be the case in a systematic review), the scoping study method is guided by a requirement to identify all relevant literature regardless of study design”. 

Point 2: Looking at the list of included review studies, there are a lot of variation in the relationships between the built environment features and behaviour (PA/diet). I was initially expected to see how this study would deconstruct the black box of different built environment features (recreation facilities, parks, trails etc) on behaviour but based on the conceptual model (fig3), built environment features are clumped together as ‘external built environment’. How do you develop theme/code in Nvivo if one of the built environment of interest was not associated with physical activity?

Response 2: The aim of this paper was to deconstruct interactions that had already been validated by the systematic reviews identified through the systematic online search. Further, the evidence coded during the qualitative analysis was not weighted according to the occurrence of themes, nor the quality of papers appraised, since the principal objective of this scoping review was to understand the mechanism of the interactions at play by deconstructing them. As a result, findings mentioned only by few reviews were considered as relevant if they were highlighting a new dimension of an interaction between the built environment and behaviors. Hence the aim of this paper was not to discover new interactions, but to try to further understand the one validated by the literature. If there was no interaction found, then we did not deconstruct the alleged interaction. If the same feature was found to be associated to different outcomes, we then analyzed the context in order to find out all the dimensions that were found to play a role in the interaction. When repeating this process with all the coded review papers, we were eventually able to isolate 7 dimensions which were found to play a role in the interactions identified by the reviews: personal factors, external built environment, capability, internal built environment, opportunity, motivation and behavior. Hence, conceptualizing the different interaction pathways between one or several of the 7 identified dimensions was critical to understand why the same specific built environment feature did not lead to a same behavior.

Point 3: How does the author assign each arrow in the mode, why are some bi-directional arrow and some unidirectional? Was it based on the findings of the included studies? The author stated that ‘…interrupted line because the process of self-selection is not systematic and is influenced by economic, social and work-related constraints.’ However, I am still not sure if I understand the meaning of interrupted lines. I believe what was stated in the interrupted line may also be applicable to some other lines in the conceptual model not just from (1) to (2). For example, (3) capability of individual can self-select themselves into (2) external built environment and (5) opportunity. However, if these arrows are assigned based on the findings from the included articles, I would like to see a clearer explanation somewhere (in text).

Response 3: Most of the time, the external built environment was found to be imposed to individuals who did not have their say in the way it was designed. In some studies, the process of self-selection was found to moderate this. If individuals do not always have their say in the way the external built environment is designed, they can nonetheless choose to live, work, recreate in built environments that reflect their preferences in terms of facilities’ availability. The interrupted line between personal factors and the external built environment represents this specific relationship of self-selection; whereas all the solid arrows represent a combination or a multitude of possible relationships. 

Besides, for all the other relationships, the solid arrows mean that one dimension directly shapes the other one. For example, personal factors contribute to directly shape individuals’ capabilities. On the other hand, we found that personal factors influence the external built environment in an indirect way. In fact, they influence individuals’ choices of neighborhood and this choice influences the type of built environment characteristics available to them. The reviews we coded focused on situations in which individuals were choosing between already existing types of built environment characteristics and features but were not directly shaping or building their built environment. Personal factors at the population level may influence the making of public policies and have a direct impact on the way the built environment is built, however, this goes beyond the scope of this paper. Hence, demonstrating that the interrupted line between personal factors and the external built environment is a solid line needs to be done through future studies.  

Capability does indeed impact self-selection, but we did not choose to put a direct arrow from capability to the external built environment because we intended to show the complexity of the pathways linking capability to the external built environment. In fact, capability impacts individuals’ motivation, both directly and indirectly, by impacting the perceived built environment and the perceived opportunity which lead individuals to healthy or unhealthy behaviors. Those behaviors then contribute to shape individuals’ personal factors such as preferences which have a direct impact on the process of self-selection. Hence, capability does influence the process of self-selection and eventually, the external built environment, but we tried to deconstruct the different pathways through which this interaction was occurring in order to better understand it.

The arrows between motivation and opportunity, on the one hand, and between motivation and capability, on the other hand, are bidirectional because motivation was found to directly impact capability and opportunity based on the COM-B model developed by Michie et al., 2011 which served as a basis for figure 3 (Michie S, van Stralen MM, West R. The behaviour change wheel: A new method for characterising and designing behaviour change interventions. Implement Sci. 2011;6.).

Point 4: Are the sub categories (e.g. individual and social from (1) personal factors) stem from the included studies?  There seems to be a lot of overlap between components of the sub-categories. For example, there are overlap between personal factors and capability; as well as external built environment and opportunity.  To give a more specific example, (1) personal factors which included ‘social-socioeconomic status’ as the sub-category can potentially overlap with (3) capability that included ‘educational, economic’. I would like to know how the author distinguish between the two.

Response 4:

-       The sub categories for (2) External built environment and (4) Internal built environment are adapted from the model developed by Turner et al., 2018 based on the analysis of the included studies (Turner C, Aggarwal A, Walls H, Herforth A, Drewnowski A, Coates J, et al. Concepts and critical perspectives for food environment research : A global framework with implications for action in low- and middle-income countries. 2018;18:93–101).

-       The sub-categories for (3) Capability, (5) Opportunity and (6) Motivation, were adapted from the COM-B model developed by Michie et al., 2011 based on the analysis of the included studies (Michie S, van Stralen MM, West R. The behaviour change wheel: A new method for characterising and designing behaviour change interventions. Implement Sci. 2011;6.).

-       Finally, the sub-categories of (1) Personal factors were identified based on the analysis of the included studies and the general understanding of personal factors that was stemming from several socioecological models which inspired the conceptual model developed in this paper and were cited in the discussion section. These models are:

·      The Social-Ecological Model for Food & Physical Activity Decisions developed by the Centers for Disease Control and Prevention, USA (Allen C, Ammerman A, Arline S, Brown D, Macgowan C, Mccarthy W. Health Equity Resource Toolkit for State Practitioners Addressing Obesity Disparities [Internet]. Centers Dis. Control Prev. Natl. Cent. Chronic Dis. Prev. Heal. Promot. Div. Nutr. Phys. Act. Obes. 2017);

·      Turner et al., 2018;

·      Sallis JF, Cervero RB, Ascher W, Henderson KA, Kraft MK, Kerr J. an Ecological Approach To Creating Active Living Communities. Annu Rev Public Health [Internet]. 2006;27:297–322;

·      Allen C, Ammerman A, Arline S, Brown D, Macgowan C, Mccarthy W. Health Equity Resource Toolkit for State Practitioners Addressing Obesity Disparities [Internet]. Centers Dis. Control Prev. Natl. Cent. Chronic Dis. Prev. Heal. Promot. Div. Nutr. Phys. Act. Obes. 2017; and

·      Papas MA, Alberg AJ, Ewing R, Helzlsouer KJ, Gary TL, Klassen AC. The built environment and obesity. Epidemiol Rev. 2007;29:129–43.

The model we propose make a distinction between personal factors (1) that included ‘social-socioeconomic status’ as the sub-category and (3) capability that included ‘educational, economic’. Individuals with a high social-economic status can still face hardship in accessing sport facilities (external built environment) if these facilities are expensive and far away from their home. Whereas individuals with the same personal factors (e.g.: same level of education and revenues) living closer to sport facilities that happened to also be cheaper, will have higher economic capability to access these facilities. However, although the conceptual model proposed is an attempt at identifying and defining distinct dimensions interacting, the authors acknowledge that it would be impossible to completely eliminate all the overlaps between the 7 dimensions.

Point 5: I believe the author’s statement about socioecological model is inaccurate. A core principle of ecological models is that influence from multiple levels can interact and exert synergistic effects on behaviour. Papers below justified my statement:

·      Giles-Corti B, Timperio A, Bull F, Pikora T. Understanding physical activity environmental correlates: Increased specificity for ecological models. Exerc Sport Sci Rev. 2005;33:175-181.

·      Hovell MF, Wahlgren DR, Gehrman C. The behavioral ecological model: Integrating public health and behavioral science. In: DiClemente RJ, Crosby R, Kegler M, eds. New and emerging theories in health promotion practice & research. San Francisco: Jossey-Bass Inc; 2002.

·      Ding, D., Sallis, J.F., Conway, T.L., Saelens, B.E., Frank, L.D., Cain, K.L. and Slymen, D.J., 2012. Interactive effects of built environment and psychosocial attributes on physical activity: a test of ecological models. Annals of Behavioral Medicine, 44(3), pp.365-374.

Response 5: The authors agree with the reviewer’s statement that a core principle of ecological models is that influence from multiple levels can interact and exert synergistic effects on behavior. Hence, we are not sure about which statement in the paper is going against this understanding of socioecological models.

In order to best answer the reviewer’s comment, we tried to explain point by point each part of the paper referring to socioecological models:

[Lines 28-29]: Here, the authors underlined the role of socioecological models in bringing out the multiple interactions occurring between environmental and personal factors, suggesting the existence of multiple layers of influences.

[Lines 751-766]: Here, the authors acknowledge once again the fact that influence from multiple levels can interact and exert synergistic effects on behavior. We then took the example of the Social-Ecological Model for Food & Physical Activity Decisions developed by the Centers for Disease Control and Prevention (CDC), USA and pointed out that this particular model did not elucidate the interactions’ pathways linking the different layers of influence identified. Hence, the second part of the statement was not meant to be generalized to all the existing socioecological models but only to the one developed by the CDC.

[Lines 784-789]: Although socioecological models bring out the idea of multiple layers of influences, they mainly focus on how these influences are impacting individuals and no deconstruction is made of the mechanisms by which individuals are also shaping their environment.

Point 6: Regarding the systematic search of articles, why didn’t the author include articles published in 2018? There are a few reviews relevant to built environment and PA/diet during 2018-2019. For example,

Stappers, N.E.H., Van Kann, D.H.H., Ettema, D., De Vries, N.K. and Kremers, S.P.J., 2018. The effect of infrastructural changes in the built environment on physical activity, active transportation and sedentary behavior–A systematic review. Health & place, 53, pp.135-149.

MacMillan, F., George, E., Feng, X., Merom, D., Bennie, A., Cook, A., Sanders, T., Dwyer, G., Pang, B., Guagliano, J. and Kolt, G., 2018. Do natural experiments of changes in neighborhood built environment impact physical activity and diet? A systematic review. International journal of environmental research and public health, 15(2), p.217.

Ding, D., Nguyen, B., Learnihan, V., Bauman, A.E., Davey, R., Jalaludin, B. and Gebel, K., 2018. Moving to an active lifestyle? A systematic review of the effects of residential relocation on walking, physical activity and travel behaviour. Br J Sports Med, 52(12), pp.789-799.

Rhodes, R.E., Saelens B.E., Sauvage-Mar, C., 2018. Understanding physical activity through interactions between the built environment and social cognition: a systematic review. Sports Med, 48(8), pp. 1893-1912.

Response 6: The paper was written between January 2018 and August 2018. The systematic search was initially conducted in February 2018. Hence, December 2017 was initially chosen as the limit date. We addressed the reviewer’s concern and updated our review. We ran the same search strategy on 18 March 2019 in order to retrieve papers published after 2017. The results of this search can be found in the Additional File 1, Supplementary table 6. All 4 of the above-mentioned reviews published in 2018 met the inclusion criteria and were included to the updated review. In total, the search for papers published between 2017 and 2019 allowed us to include 17 additional records that met the inclusion criteria. The methods and results sections were changed accordingly.

Reviewer 2 Report

The authors provide an interesting review of the observed interactions between built environment features and health behaviours. I enjoyed the detail provided in the appendices to the review, and provision of detailed search strategies and summary tables of included results. I would recommend major revisions, and addressing the serious flaws of the review's approach to the analysis of included papers before considering publication. I believe the paper would benefit immensely from proofreading for minor grammar and subject-verb agreement issues, and to clarify the narrow focus of the review on physical activity and dietary behaviours. I have provided more detailed comments by line numbers below.

53-58: I found the black box analogy to be helpful, but I remain concerned about the certainty with which the authors purport to be able to open up and examine the contents of this black box. I would like to see a more robust discussion in the introduction, and inclusion in the discussion, of the limitations in attempting to construct an effects-interaction model that aims to encapsulate environmental effects on behaviours. I worry the authors are treading into unfounded causality, rather than simple observational correlation.

73: I do not find Figure 1 helpful in advancing our understanding of the relationship between built environment and health-related behaviours. How does each built environment variable relate to physical activity and dietary variables independently to BE features from a conceptual perspective? This figure should be your hypothesis about the relationship you are investigating through the review.

132-168: I find the manner in which the review evidence was approached through subsetting to be quite problematic. It remains unclear why the subsetting occurred from the authors' description, and appears to be more 'convenience' than a defensible approach. Clarification here would be integral to supporting publication of the review. How do you distinguish between coded papers, and narrative-only papers?

206: Table 2 would make a larger contribution to the field if presented in more graphical format by population, study design, and outcome type.

209-267: I appreciate the authors attempt to construct a pathway model of built environment influences and health behaviours. However, I found the graphic to be unclear and confusing, and the surrounding descriptive language to be overly verbose. I would suggest the authors refer to Figure 1 in Gomez SL etal. The impact of neighborhood social and built environment factors across the cancer continuum: current research, methodological considerations, and future directions. Cancer. 2015 Jul 15;121(14):2314-30 or Figures 1 and 2 in Stafford M etal. Pathways to obesity: identifying local, modifiable determinants of physical activity and diet. Social science & medicine. 2007 Nov 1;65(9):1882-97, for inspiration on how to structure their figures and description of the model.

332: I believe the discussion section could be more robust by breaking the content into relevant subheadings, and providing recommendations targeted to the realms of the academic, policy-makers, and public. 

447-451: I found the assertion that this paper provides an authoritative model of health effects from built environment to be problematic. I would encourage the authors to be more cautious with their language, and describe how their hypothesized relationships from the background literature is different from their observed findings in the review. A conceptual model can only be presented as such after rigorous debate among scholars. Instead, it would be more valuable, and likely worthy of debate, for the authors to present their findings as a graphical representation of the theoretical and captured research findings that link built environment features with physical activity and dietary behaviours.

Author Response

Response to Reviewer 2 Comments

Remarks:

-       We updated the review and ran the same search strategies in the same databases in order to include all relevant papers published between 2017 and 2019.

-       Changes in the supplementary files and tables were highlighted in yellow.

-       Changes in the manuscript can be seen in track change and were copied in bold and darker red when integrated to the responses bellow.

General comment: I believe the paper would benefit immensely from proofreading for minor grammar and subject-verb agreement issues, and to clarify the narrow focus of the review on physical activity and dietary behaviours.

Response to general comment: The paper has been proof-read by native speakers in order to correct the remaining grammar and subject-verb agreement issues. Corrections have directly been made in the text of the manuscript.

Physical activity and dietary behaviors are very often studied together since they are major risk factors of obesity and chronic diseases such as Type 2 diabetes. Further, both physical activity and dietary behaviors are in themselves very broad topics. For all these reasons, it appeared relevant to the authors to first focus on these two types of behaviors in order to develop a conceptual model that could then be tested by future studies and applied to other types of behaviors.

Point 1: [ll. 53-58]: I found the black box analogy to be helpful, but I remain concerned about the certainty with which the authors purport to be able to open up and examine the contents of this black box. I would like to see a more robust discussion in the introduction, and inclusion in the discussion, of the limitations in attempting to construct an effects-interaction model that aims to encapsulate environmental effects on behaviours. I worry the authors are treading into unfounded causality, rather than simple observational correlation.

Response 1: The introduction section has been revised and the aim of the study clarified [lines 28-58]. The changes made specify that the aim of this scoping review was not to examine causal relationships but rather to bring out the complexity of the interactions’ pathways. In order to do so, we tried to deconstruct the black box of interactions by identifying interactions’ pathways linking the 7 following dimensions: personal factors, external built environment, capability, internal built environment, opportunity, motivation and behavior. These dimensions and the way they interplay to create interactions between the built environment and PA and/or diet, are grounded in the evidence that emerged from the systematic literature search. Hence, this paper aimed at conceptualizing these interactions ‘pathways and future research should try to validate the model by testing it on interactions between specific built environment features and specific health related behaviors.

Further, [lines 782-783], the authors clearly stated that the conceptual model developed “does not aim at drawing causality links or systematic associations but at understanding key mechanisms so that the relationship between behaviors and the built environment can be analyzed in different contexts”.

Finally, causality was also tackled in the discussion part of the manuscript, [lines 820-822]: “Referring to the mixed results alluded to in the discussion, the present study is limited by the studies included in the reviews. Several reviews highlighted the limitation of the cross-sectional study design often used in the primary studies [24,50,63]. In fact, cross sectional studies do not allow to draw any conclusion about causality”.

Point 2: [l. 73]: I do not find Figure 1 helpful in advancing our understanding of the relationship between built environment and health-related behaviours. How does each built environment variable relate to physical activity and dietary variables independently to BE features from a conceptual perspective? This figure should be your hypothesis about the relationship you are investigating through the review.

Response 2: Figure 1 does not aim at advancing the reader’s understanding of the relationship between the built environment and health related behaviors but rather to conceptualize what the authors mean by “black box of interactions”. The aim of this figure was to show to the reader that unknown factors were interacting and were playing a role in the relationship between the built environment behaviors pertaining to PA and/or diet. This figure was meant to show the reader what the focus of this paper was and was placed at the very beginning of the paper in the paragraph about “stage 1: Identifying the research questions”. The black box is an analogy of what we are looking for and Figure 3 (the conceptual model) describes what we found behind the black box. We addressed the reviewer’s comment by revising the introduction in order to make the aim of the black box analogy clearer but decided to keep figure 1 based on other reviewers’ comments.

Point 3: [ll. 132-168]: I find the manner in which the review evidence was approached through subsetting to be quite problematic. It remains unclear why the subsetting occurred from the authors' description, and appears to be more 'convenience' than a defensible approach. Clarification here would be integral to supporting publication of the review. How do you distinguish between coded papers, and narrative-only papers?

Response 3: We would like to take the opportunity to further clarify the rationale to our approach. Once the screening process was completed, it was apparent that many reviews had the same focus and conclusions. The concept of saturation (the point in coding when you find that no new codes occur in the data), which is grounded in the methodology developed in Glaser, BG. Theoretical sensitivity: Advances in the methodology of grounded theory. Mill Valley, CA: Sociology Press. 1978, was applied to our dataset.

All papers bringing new evidence were coded in order to develop new themes and have a complete overview of the different dimensions of the interactions identified in the literature. The total number of paper coded reached the number of 45; leaving 45 papers which did not bring any new coding themes because the evidence presented was already covered by one or several of the papers already coded. Hence, all 90 papers were thoroughly read and were used for the narrative analysis. The conceptual model was drafted based on the themes developed through the coding of the above-mentioned 45 papers (Subset 1), knowing that these themes were covering all the evidence mentioned in the 45 other papers (Subset 2).

The authors feel the accumulation of identical coded material would not have contributed to the narrative analysis nor to the development of the conceptual model.

Point 4: [l.206] Table 2 would make a larger contribution to the field if presented in more graphical format by population, study design, and outcome type.

Response 4: The reader can find all the above-mentioned information in the Additional files. Additional File 3, “Supplementary Table 1 – Included records: description of Subset 1 and Subset 2” covers all the following dimensions: population, study design and outcome type. It presents in a graphical format, the references of the included records, the focus (PA; Diet; PA and Diet), the date range of included studies, the aim & objectives, the population, the methodology and key findings. Since it includes 90 papers, the authors felt this table was too large (14 pages) to be directly included in the body of the manuscript. Hence, we decided to openly provide it to readers as an additional file (Cf. Additional File 3.). However, we also created Table 3 which aggregates the information from Supplementary Table 1, Additional File 3. Table 3 can be found directly in the manuscript.

Regarding Table 2, its aim was not to present studies by population, study design and outcome type – since it is already done in Table 3 and in Supplementary Table 1, Additional file 3 – but to develop the thematic framework used for coding subset 1. Hence it is not part of the “Results” section but of the “Materials and Methods” one. 

The rationale for the development of Table 2 can be found in the manuscript [lines 825-829]. For more clarity, the development process of Table 2 was also summarized as follows in the “Materials and Methods” section.

Point 5: [ll.209-267]: I appreciate the authors attempt to construct a pathway model of built environment influences and health behaviours. However, I found the graphic to be unclear and confusing, and the surrounding descriptive language to be overly verbose. I would suggest the authors refer to Figure 1 in Gomez SL etal. The impact of neighborhood social and built environment factors across the cancer continuum: current research, methodological considerations, and future directions. Cancer. 2015 Jul 15;121(14):2314-30 or Figures 1 and 2 in Stafford M etal. Pathways to obesity: identifying local, modifiable determinants of physical activity and diet. Social science & medicine. 2007 Nov 1;65(9):1882-97, for inspiration on how to structure their figures and description of the model.

Response 5: Although Figure 1 in Gomez SL et al. (2015) gives a very good and graphic example of the impact of social and built environment characteristics on the cancer continuum, it does not aim at deconstructing the black box of interactions as we attempted to do in our model. It provides a list of built and social environment factors which are playing a role in the cancer continuum without explaining the way individuals are interacting with their environment. For example, it does not take into account factors such as capabilities, motivation or individuals’ perception of opportunities.

Regarding figures 1 and 2 in Stafford M et al. (2007), that would be the next step to validate our conceptual model. In fact, they are looking at very narrow interactions when we are more interested in the qualitative characteristics of the interactions between the built environment and PA and/or diet related behaviors.  

However, we reorganized the text describing Figure 3 – as suggested – in order to give a more graphic and a clearer explanation of the conceptual model.

Point 6: [l.332]: I believe the discussion section could be more robust by breaking the content into relevant subheadings, and providing recommendations targeted to the realms of the academic, policy-makers, and public.

Response 6: The aim of this paper was to provide a methodological tool that will help academic, policy makers and the public to deconstruct the interactions and help them better understand the way individuals interact with their built environment. Hence, we rather talked of “key implications” than “recommendations”. We developed the key implications of our conceptual model in the “Discussion” section [lines 791-799]. Additionally, we also expanded the conclusions in order to give more specific examples of how our new model might help research and practice [lines 839-853].

Point 7: [ll.447-451]: I found the assertion that this paper provides an authoritative model of health effects from built environment to be problematic. I would encourage the authors to be more cautious with their language, and describe how their hypothesized relationships from the background literature is different from their observed findings in the review. A conceptual model can only be presented as such after rigorous debate among scholars. Instead, it would be more valuable, and likely worthy of debate, for the authors to present their findings as a graphical representation of the theoretical and captured research findings that link built environment features with physical activity and dietary behaviours.

Response 7: It was not the authors intention to provide an authoritative model of health effects from the built environment. We revised the introduction as follows in order to make the aim of the paper clearer and to insist on its conceptual and non-authoritative dimension [lines 48-58].

Reviewer 3 Report

Overall the paper is well written and I believe will be of interest to readers. Specific and general comments are below:

1) line 50: "...And why does the availability of of such facilities not necessarily lead to the same behavior in different people?"  This question can be interpreted as being naive in my opinion. Human behavior has been a mystery for a long time and will probably be for a long time. 

2) page 169   in Table 2: Do you mean "chronic diseases" or "chronicle" as you have written.

Figure 2 : It indicates 835 records here and 832 on the previous page, clarify.

3) line 216-217 - it should not be difficult but I think that it would help to clarify reader understanding by providing and example of how behavior is a source of behavior.

4) As well as the paper is written the reference list and Supplementary Table one need some work. Inconsistency of citation styles and odd use of CAPS. For instance Seattle and King County are not capitalized in reference #10.

Author Response

Response to Reviewer 3 Comments

Remarks:

-       We updated the review and ran the same search strategies in the same databases in order to include all relevant papers published between 2017 and 2019.

-       Changes in the supplementary files and tables were highlighted in yellow.

-       Changes in the manuscript can be seen in track change and were copied in bold and darker red when integrated to the responses bellow.

Point 1: [line 50]: "...And why does the availability of of such facilities not necessarily lead to the same behavior in different people?"  This question can be interpreted as being naive in my opinion. Human behavior has been a mystery for a long time and will probably be for a long time. 

Response 1: We have reworded the text to address a larger audience who would not be familiar with behavioral sciences and in order to debunk pre-conceived ideas about behaviors. We tried to make the sentence sound less naive by modifying lines [41-47].

Point 2: [line 169] in Table 2: Do you mean "chronic diseases" or "chronicle" as you have written.

Response 2: Yes, we did mean “chronic diseases”. The correction has been made directly in the text of the manuscript (line [371], Table 2).

Point 3: Figure 2: It indicates 835 records here and 832 on the previous page, clarify.

Response 3: 835 was the total number of records retrieved from the electronic search (832) plus the total number of records retrieved through the manual search of the Active Living database (3). It was also the number that appears at the top of the PRISMA flowchart. However, we updated the search by running the search strategies for the year span from 2017 to 2019. Hence, the lines [394-397] were clarified.

Point 4: line [216-217] - it should not be difficult but I think that it would help to clarify reader understanding by providing and example of how behavior is a source of behavior.

Response 4: Examples and clarifications of how behavior is a source of behavior are given further down in the manuscript. The following lines [555-556] were aimed at clarifying the reader’s understanding of how behavior is a source of behaviour. Additionally, we clarified the lines [708-715] in order to give

Point 5: As well as the paper is written the reference list and Supplementary Table one need some work. Inconsistency of citation styles and odd use of CAPS. For instance Seattle and King County are not capitalized in reference #10.

Response 5: The reference lists of the manuscript and of all Supplementary files and included tables were checked and odd uses of CAPS were corrected. Citation styles were harmonized in order to ensure consistency.

Reviewer 4 Report

This review contributes to the growing body of literature surrounding the association between the built environment and PA/diet. The scoping review methodology was thoroughly described and relevant. The model is a bit difficult to interpret, however the use of corresponding numbers/letters helps the reader. Overall, the proposed shift from external to internal influences is significant and interesting. 

Author Response

Response to Reviewer 4 Comments

Remarks:

-       We updated the review and ran the same search strategies in the same databases in order to include all relevant papers published between 2017 and 2019.

-       Changes in the supplementary files and tables were highlighted in yellow.

-       Changes in the manuscript can be seen in track change and were copied in bold and darker red when integrated to the responses bellow.

General comment: The model is a bit difficult to interpret, however the use of corresponding numbers/letters helps the reader.

Response to general comment: [lines 554-556] we reorganized the text describing Figure 3 in order to give a more graphic and a clearer explanation of the conceptual model.

Point 1: I expected concrete research questions to be presented here. Including the phrase ‘such as’ leaves the reader questioning the framework and aim of the study.

Response 1: The introduction section has been revised and the aim of the study clarified [lines 28-58]. The changes made specify that the aim of this scoping review was not to examine causal relationships but rather to bring out the complexity of the interactions’ pathways. In order to do so, we tried to deconstruct the black box of interactions by identifying interactions’ pathways linking the 7 following dimensions: personal factors, external built environment, capability, internal built environment, opportunity, motivation and behavior. These dimensions and the way they interplay to create interactions between the built environment and PA and/or diet, are grounded in the evidence that emerged from the systematic literature search. Hence, this paper aimed at conceptualizing these interactions ‘pathways and future research should try to validate the model by testing it on interactions between specific built environment features and specific health related behaviors.

Point 2: Did you also try including ‘walking’? Much of the literature on the built environment-behavior relationship focuses on walking specifically.

Response 2: Yes, we did include ‘walking’ as a key word in the search strategy. It was listed in Table 1 as “walking” and “walk*” in order to retrieve all words beginning by “walk” and make sure to retrieve as many variations of the word as possible. Further, all the keywords combinations are detailed in Additional File 1. [Lines 185-190] we clarified the text of the manuscript in order to make it clearer.

Point 3: Did you also try seraching with food?

Response 3: Yes, we did include ‘food’ as a key word in the search strategy. It was listed in Table 1 as “food”, “unhealthy food”, “unhealthy foods”, “food choice”, “food choices” and “food decisions”. All these variations were used in the online search strategy. Further, all the keywords combinations are detailed in Additional File 1. [Lines 185-190] we clarified the text of the manuscript in order to make it clearer.

Round 2

Reviewer 1 Report

The author described Black box as “the unknown mechanisms at play…”. However, from what I read, the aim of this paper was to give an overview of research findings linking built environment with PA and diet behaviours—which is something that is known from the literature. Theoretically, assuming that the definition of black box stated by the author is accurate, this paper is not deconstructing a black box (the unknown mechanisms), but rather summarising current (known) evidence. Therefore, I am having doubts on the use of the term ‘black box’ in the title as well as throughout the article.  

In the abstract, the author stated “Most review papers confirmed the influence of the built 14 environment on the behaviors of interest with some noting that a same built environment feature 15 could have different behavioral outcomes. The conceptual model developed sheds light on these 16 mixed results and brings out the role of several personal and behavioral factors in the shift from the 17 measured to the perceived built environment. This shift was found to critically shape individuals’ 18 behaviors and to have the power of redefining the strength of every interaction.”

I don’t think the author has shed light on the mixed results of the built environment feature in different behaviour outcomes in this paper.

I think the biggest concern I have for this paper is that the conceptual model lacks specificity and it is confusing. Going back to my comment 2, I think clumping any built environment feature into ‘external built environment’ lacks specificity. There are a lot of variation within the built environment and PA/diet in the literature. Also, combining PA and diet into one group (as shown in Fig 3), can be problematic. Some features may be relevant to PA but not diet, even within PA, some built environment features may operate differently for different PA domains (e.g., public transport infrastructure is positively associated with transport-related PA but not leisure-type PA; recreation facilities is positively associated with leisure-time PA but not transport-related PA). PA and diet should not be view together as the influence of certain built environment may have independent effect on PA and diet.

I am confused by the statement that described the ‘b’ pathway in the conceptual model (on page 12; table number not labelled)—‘educational and economic capabilities are impacted by socioeconomic factors’. Based on my understanding, educational and economic capabilities is the same as socioeconomic factors (usually comprised of education, occupation and income). Can the author please clarify?

Author Response

Response to Reviewer 1 Comments (Round 2)

Comment 1: The author described Black box as “the unknown mechanisms at play…”. However, from what I read, the aim of this paper was to give an overview of research findings linking built environment with PA and diet behaviours—which is something that is known from the literature. Theoretically, assuming that the definition of black box stated by the author is accurate, this paper is not deconstructing a black box (the unknown mechanisms), but rather summarising current (known) evidence. Therefore, I am having doubts on the use of the term ‘black box’ in the title as well as throughout the article. 

Response to comment 1: We appreciate the reviewer’s comment and would like to take the opportunity to further clarify and motivate why we think it is appropriate to use the black box analogy. The understanding of a black box is based on the hypothesis of a relation between the input (e.g. built environment like more bicycle lanes) and the output (e.g. increased physical activity). We refer to the “between” portion as the interaction. As stated in lines 53-56, the authors aimed to synthesize the current known evidence into one comprehensive model that then would be used elucidate why certain interventions worked in one setting and not in another. The literature review enabled us to (1) identify all the relevant components of the interactions of interest (= synthesis step) and (2) then to analyze the different pathways in order to bring out the links and relationships between these components (= deconstruction phase). In other words, the authors constructed a model that would allow a researcher to take any given set of combinations (i.e. built environment, behavior, context) and outline the possible pathways and how they are related. This process is what we call "deconstructing the black box of interactions”. Given that our use of the black box analogy is accurate and that it has been used to describe similar examples in published public health and implementation science literature (Salter & Kothari 2014, Nielson & Randall 2013), the authors strongly feel justified in our use of the black box analogy in the title and text of the article. 

Comments 2 & 3: In the abstract, the author stated “Most review papers confirmed the influence of the built environment on the behaviors of interest with some noting that a same built environment feature could have different behavioral outcomes. The conceptual model developed sheds light on these mixed results and brings out the role of several personal and behavioral factors in the shift from the measured to the perceived built environment. This shift was found to critically shape individuals’ behaviors and to have the power of redefining the strength of every interaction.” I don’t think the author has shed light on the mixed results of the built environment feature in different behaviour outcomes in this paper.

I think the biggest concern I have for this paper is that the conceptual model lacks specificity and it is confusing. Going back to my comment 2, I think clumping any built environment feature into ‘external built environment’ lacks specificity. There are a lot of variation within the built environment and PA/diet in the literature. Also, combining PA and diet into one group (as shown in Fig 3), can be problematic. Some features may be relevant to PA but not diet, even within PA, some built environment features may operate differently for different PA domains (e.g., public transport infrastructure is positively associated with transport-related PA but not leisure-type PA; recreation facilities is positively associated with leisure-time PA but not transport-related PA). PA and diet should not be view together as the influence of certain built environment may have independent effect on PA and diet.

Response to comments 2 & 3: The authors revised the manuscript for clarity (line 94-96). The added sentence, aims at better explaining why diet and PA were tackled together. From a public health standpoint, diet and PA are complementary behaviors and often need to be addressed in combination. This was our rationale for combining the two. This does not exclude the possibility of using the model to understand the interactions related to either diet or PA behaviors separately. With respect to the lack of specificity, the aim of the paper is to focus on the mechanisms that drive the interactions between BE and behavior (diet and PA). Hence, it cannot be more specific because every relationship depends on the context and the built environment characteristics + personal factors that are specific to each context. It is important for the reader to have an understanding of the mechanisms so that every observed interaction can be deconstructed into these seven components and specifically analyzed. We do not aim to say that a specific BE characteristic named A leads to a specific diet or PA behavior named B, because that would be incorrect. What this model offers is an understanding of the pathway, i.e., given an observed (or measured) BE characteristic named A and an observed (or measured) diet or PA behavior named B, what is the likely mechanism that has led to this behavioral outcome, B in one context but not in another. Hence, our aim is not to say which specific features lead to what specific behavior in a given context as that is not possible, given the ‘n’ number of BE characteristics, diet and PA behaviors and different contexts around the world, resulting in an infinite number of permutations and combinations. It is however possible to use this model to understand the mechanism at play in any given set of these infinite combinations. This is also the reason we refer to the mixed results in the abstract (see comment 2 above). Mixed result are also explained further in the discussion part of the manuscript (line 303-307 and 313-324). Further, the authors also revised the manuscript for more clarity about the role of context (line 354-355; 363-364; 371-373).

Comment 4: I am confused by the statement that described the ‘b’ pathway in the conceptual model (on page 12; table number not labelled)—‘educational and economic capabilities are impacted by socioeconomic factors’. Based on my understanding, educational and economic capabilities is the same as socioeconomic factors (usually comprised of education, occupation and income). Can the author please clarify?

Response to comment 4: The authors agree this could cause confusion thus revised for clarity: “Educational, economic and psychosocial capabilities are impacted by social determinants and other factors such as age and attitude” (line 230).
